# Impact of urban productive safety net program on urban households' asset accumulation and food consumption rate in Dessie City, South Wollo Zone, Amhara Region, Ethiopia

**Alem-meta Assefa** *

Department of Geography and Environmental Studies, Wollo University, Dessie, Ethiopia

* alemmeta99@yahoo.com

**Data Availability Statement:** Data is available within the paper.

## Abstract

Agriculture is the driver of economic growth in many developing countries like Ethiopia, where it often represents at least 25% of gross domestic product. But the quality and productivity of the land depends on the way agriculture is practiced. Better agricultural techniques are necessarily including Productive Safety net Program. Urban Productive Safety Net Program is designed to support the poor households and establish urban safety net mechanisms. While several studies have examined the effects of rural productive safety net programs, the outcomes of similar programs in urban areas are not well documented. This knowledge gap hinders the ability to effectively design and implement urban productive safety net programs. The main objective of this study was to assess the impact of urban productive safety net program on asset accumulation and consumption rate in Dessie city. Both primary and secondary data sources were used. A total of 112 households were selected randomly among the public work Urban Productive Safety Net Program beneficiary and non-beneficiary households. Descriptive analysis, inferential statistics and econometric models were used for analysis. Based on the econometric estimation results the demographic and socio-economic variables such as education, access to credit, access to extension services and adult male labor force revealed positive relation to public work urban productive safety net program beneficiary and non-beneficiary household's asset accumulation and consumption. The impact of program participation was also found to be positive and significant for both home asset and community asset. While food and non-food consumption even the total consumption and food security status are also positive. Generally, the result indicated that, due to program participation beneficiary households have higher home asset and community asset and better consumption and food security status.

**Funding:** The author(s) received no specific funding for this work.

**Competing interests:** The authors have declared that no competing interests exist.

## 1. Introduction

In Sub-Saharan Africa, numerous individuals continue to experience chronic food insecurity caused by repeated droughts, limited productivity in subsistence agriculture, and widespread poverty [1]. Samuel [2] states that various food security initiatives have been created to address this ongoing issue effectively. Accordingly, in the past two decades we have seen a rapid increase of social protection programs in African countries to alleviate poverty, food insecurity, and vulnerability of poor households [3]. These formal programs aim to assist individuals and households facing chronic food insecurity or temporary financial setbacks by offering income support or alternatives, such as cash and in-kind transfers, subsidies, and labor-intensive public work. The Productive Safety Net Program (PSNP) is a large-scale social protection program implemented in Ethiopia since 2005. It aims to provide assistance to chronically food-insecure households and help them improve their food security and resilience to shocks. The program combines two main components: direct support through cash and/or food transfers to eligible households, and support for public works programs. The cash and food transfers are intended to meet immediate consumption needs, while public works programs provide temporary employment opportunities and help build community assets such as roads, bridges, and irrigation systems.

The relative lack of change in hunger at the global level from 2021 to 2022 hides substantial differences at the regional level. Progress was made towards reducing hunger in most sub regions in Asia and in Latin America, but hunger is still on the rise in Western Asia, the Caribbean and all sub regions of Africa. The proportion of the population facing hunger is much larger in Africa compared with the other regions of the World –nearly 20 percent compared with 8.5 percent in Asia, 6.5 percent in Latin America and the Caribbean, and 7.0 percent in Oceania [4].

An estimated 149 million Africans are facing acute food insecurity—an increase of 12 million people from a year ago [5]. This equates to a risk category of 3 or higher (Crisis, Emergency, and Catastrophe) on the Integrated Food Security Phase Classification (IPC) scale of 1 to 5. Some 122 million of those facing acute food insecurity are in countries experiencing conflict—82 percent of the total—accentuating that conflict is the primary driver of acute food insecurity in Africa [6]. Four of the top 10 countries facing the most acute food insecurity are in East Africa—Sudan, Ethiopia, South Sudan, and Somalia. Ethiopia, in particular, had an estimated 20.1 million people who are facing food insecurity, and 15.1 million require emergency food assistance, following a 2-year war in Tigray, conflicts in other parts of the country, as well as a severe drought that struck the Horn of Africa [6]. These individuals lacked consistent access to sufficient food, leading to asset selling and consumption restrictions in order to survive [7].

Before implementing PSNP, the Ethiopian government relied on unpredictable annual appeals for emergency food relief, which saved lives but failed to address the increasing number of food insecure individuals [7]. Based on the information provided, the study conducted by Berhane et al. [8] indicates that the second phase of the Productive Safety Net Program (PSNP) in Ethiopia has several successes when compared with the initial emergency relief appeal method. These successes include:

**Enhanced food security**: The study suggests that the PSNP plays a crucial role in ensuring food security for both chronically and temporarily food insecure households. By providing program transfers and supporting productive activities, the PSNP aims to assist over 7.5 million people in 318 districts across eight regions. This targeted approach is likely more effective in addressing the needs of food insecure individuals compared to the unpredictable annual appeals for emergency food relief [9].

**Long-term planning and stability:** Unlike the initial emergency relief appeal method, which relied on unpredictable annual appeals for food aid, the PSNP operates on a set timeline until December 2014. This longer-term planning allows for more stability and predictability in addressing food insecurity issues [10].

**Diversification of support:** The PSNP not only provides program transfers but also supports productive activities. This indicates that the program aims to go beyond providing immediate relief by helping households develop sustainable livelihoods and improve their resilience to food insecurity.

However, without further information on the specific failures of the initial emergency relief appeal method, it is challenging to make a direct comparison. It is important to note that the PSNP is a complex program with its own limitations and challenges. Further analysis and evaluation would be needed to understand the full extent of its successes and failures in comparison to the initial emergency relief appeal method.

In an attempt to break the cycle of annual appeal of food aid and achieve an acceptable level of food security at macro (national) and micro (household) level, the government of Ethiopia developed a food security strategy (FSS) in November 1996 [11]. The FSS highlighted government plans to address the causes and effects of food insecurity in Ethiopia. Based on the FSS, the government designed regional food security programs and projects in 2002 [12].

The PSNP is currently the largest operating social protection program in Sub-Saharan Africa, outside of South Africa. It differs from previous food-for-work programs, in that it focuses continuously on selected households over several years and in that the explicit objective was that it will eventually be phased out [13].

The Ethiopian government has been working on a program called the Urban Productive Safety Net Program (UPSNP) that will span a 10-year period. This program is part of the Urban Food Security and Job Creation Strategy, which was officially endorsed on May 8, 2015. The purpose of the UPSNP is to provide assistance to more than 4.7 million impoverished individuals living in 972 cities and towns across the country. It is envisaged that this will be achieved over a long-term period through a gradual roll-out plan of different phases starting with big cities having a population of over 100,000 people. The proposed Bank support will provide assistance for the first five-year phase of the government program and is targeting 11 major cities, Addis Ababa and one city from each region (Adama, Assayita, Asosa, Dessie, Dire Dawa, Gambella, Hawassa, Harari, Jijiga, and Mekele). In the initial stage of the UPSNP, approximately 604,000 individuals who are among the most impoverished 12 percent and around 55 percent of the population residing below the poverty threshold in these 11 cities were the recipients of the program's benefits [14].

Dessie City was one of chronically food insecure cities in Ethiopia. The first phase of the program started in 2017 through direct support and public work by choosing beneficiaries from selected *Kebeles* [15]. Despite the introduction of multiple food security interventions with PSNP, Dessie City has been classified as a typical food insecure area. Having these in mind, the researchers tried to evaluate the impacts of urban productive safety net program on urban households' asset accumulation, consumption and food security status in Dessie City. Accordingly, the conceptual framework shows several factors; like personal, socio-economic, institutional and bio-physical factors that affect household participation in UPSNP and its impact.

Historically, urban poverty has consistently faced the significant challenge of food insecurity, especially in drought-prone areas of Ethiopia [16]. The responses to food security have conventionally been dominated by emergency food-based intervention. High proportion of food insecure households have been receiving emergency food aids. Although the emergency assistance was substantial and saved many lives, there was uncertainty of the food aid, poor

timing of provision and insufficient quantity of aid for the individual's beneficiary was among the main defects of food aids in the past and creating dependency syndrome on the recipients [17]. For that reason, the past decades of large-scale food aids deliveries have done little to prevent households' asset depletion, prevent environmental degradation and chronically food insecurity in general, because of ignorance of integrating these aids with natural resource management in the country.

Recognizing this, the government of Ethiopia changed the emergency food-based assistance to multi-year support of PSNP in 2005 that provides cash and food transfers to chronically food insecure households with the aim of preventing asset depletion at household level while creating assets at the community level [18]. At national level, there were some works that tried to highlights the impacts of program intervention, particularly PSNP. Unfortunately, most of them have focused on the impacts of PSNP on asset building and consumption in rural areas rather than urban areas. According to Kataru's [19] research finding, it has been determined that the Productive Safety Net Program (PSNP) is capable of effectively mitigating consumption fluctuations and safeguarding household assets. Berhane et al. [13] indicated that beneficiary households have managed to significantly improve their food security status by 1.05 months. On the contrary, what has been demonstrated is that there is no noteworthy distinction between the consumption and assets of individuals involved in the PSNP program and those who are not [20]. Similarly, Andersson et al. [21] indicated that there was no increment in livestock holdings for program participants. Furthermore, Habitamu [22], stated that, participation in PSNP reduces the asset size of the participants, due to the dependency behavior of beneficiaries which discourage them to put efforts on their agricultural activities. But, none of them discussed the magnitude of the impact of PSNP on asset accumulation and consumption on urban households.

From the above statements, we understand that various researchers have examined the effects of the program at both the national and local levels, but their findings have been inconsistent and far from conclusive. Moreover, while several studies have examined the effects of rural productive safety net programs, the outcomes of similar programs in urban areas are not well documented. This knowledge gap hinders the ability to effectively design and implement urban social protection interventions that can enhance the economic well-being and food security of vulnerable urban populations. Consequently, this study was conducted to address these knowledge gaps.

## 2. Materials and methods

### 2.1 Description of the study area

Dessie City is aged more than a hundred and ten years old and also capital of Dessie Zuria *Wereda* and South administrative Wollo Zone. It is located between 11˚08'North Latitude and 39˚38'East Longitude. It is also situated in the north eastern part of the country at 401km away from Addis Ababa, the capital of Ethiopia. The city is placed at the foot of *Tossa* Ridge and is surrounded by and also incorporates a number of hills within its boundary. The average elevation of the city is 2,600 m.a.s.l. Dessie was established in 1893 by King Michael. The city serves as social, economic and political center for the South Wollo Zone and their administration. According to the Central Statistics Agency [23], the total population of the city is approximately 250,346 and its annual growth rate is 3.72%. The sex composition of the population is 46.8% male and 53.2% female. The city has 5 sub-cities and its total area is 16,745.82 hectares.

### 2.2 The research design and sampling

The research was conducted through descriptive survey design. The survey design helps to identify patterns of behavior, activities, people's attitude, opinions, habits and any of the

variety of social problems that influence the explicability and scaling-up efforts of the program across various contexts [24]. The utilization of a descriptive survey design aids in identifying past occurrences and enables the pre-determination of specific preventive measures that can be implemented.

The study used mixed research approach. Accordingly, both quantitative and qualitative data were collected and analyzed at the same time. Data obtained through interview, questionnaire and observation were analyzed qualitatively. The quantitative data were analyzed and interpreted through appropriate statistical tools and cross tabulations were made to facilitate meaningful analysis and interpretations of the research findings.

**Selection of sample site.** On the bases of the researchers' prior knowledge of the area and work place Dawudo Sub-city was selected purposively among the 5 sub-cities of Dessie City. Then, in Dawudo sub-city, Robit and Bahil Amba *Kebeles* were selected by simple random sampling method.

**Selection of sample households.** Creswell [25], stated that all target populations are member of a real set of subjects to which a researcher wishes to generalize the results of the study. The study specifically focused on each chosen Kebele, and the researcher obtained the complete roster of household heads from the Kebele administration office.

Then on the bases of *Kebele* document there are 338 Public Work-Productive Safety Net Program (PW-PSNP) beneficiary households and 272 eligible non-beneficiary households in Robit *Kebele*. While 337 PW-PSNP beneficiary households and 273 eligible non-beneficiary households in Bahil Amba *Kebele*, thus totally 1220 both in PW-UPSNP beneficiary and eligible non-beneficiary household heads were used as sample frames included for the survey. Then after, the researcher used the known formula so as to minimize the number of household heads in the study as a sample. Thus, Kothari [26] sample size determination was employed as shown below:

$$n = \frac{z^2.p.q.N}{e^2(N-1) + Z^2.p.q}$$

Where: n = Sample size

z = Standard variation at 95 percent confidence interval (1.96)

p = Sample proportion in the target population estimated to have the characteristics being measured (0.03)

q = 1-p

N = size of the target population is 1220

e = the estimate should be within 3 percent of the true value (0.03)

Therefore, the desired sample is $n = \frac{1.96^2.0.03.(1-0.03).(1220)}{0.03^2(1219-1)+(1.96)^2.0.03.(1-0.03)} = 112.4$

$$n = \frac{0.03(1.96^2 \ x0.97x1220)}{0.03(0.03 \ x \ 1219+1.96^2)} = \frac{3.8416x0.97x1220}{36.57+3.8416} = n = \frac{4546.14944}{40.4116} = 112$$

$$n = \mathbf{112}$$

Then by using the above formula, 112 household heads were selected as a representative sample. These household heads were selected based on proportional random sampling technique or ratio to the total household heads (Table 1). In addition to this, interviews were conducted with six key informants, specifically selected *Kebele* Development Agents, *Kebele* Administrators, and the head of the UPSNP Task Force members. Finally, both the sample households and the key informants were orally asked to express their agreement for data collection interview. As a result, they expressed their consent verbally.

**Table 1. Sampling frame and sample size of household heads.**

| Dawudo sub city | Target population | | | Size of sample households | | |
|---|---|---|---|---|---|---|
| | PWPSNP beneficiary | Eligible-UPSNP non- beneficiary | Total | PWPSNP Beneficiary | UPSNP non- beneficiary | Total |
| *Robit Kebele* | 338 | 272 | 610 | 31 | 25 | 56 |
| *Bahilamba Kebele* | 337 | 273 | 610 | 31 | 25 | 56 |
| **Total** | 675 | 545 | 1220 | 62 | 50 | 112 |

## 2.3 Data sources and collection techniques

The study employed both primary and secondary data sources. The primary data were collected from public work UPSNP beneficiary and eligible non-beneficiary household heads through questionnaires as well as key informant interview. While secondary data were collected from different published and unpublished sources of data. Data collection techniques of the study include questionnaires, key informant interview, and field observation.

## 2.4 Methods of data analysis

The study employed both quantitative and qualitative data analysis techniques. Quantitative data were gathered through the close-ended questionnaire and analyzed by using descriptive statistics, inferential statistics and econometric tools using stata computer software. Similarly, qualitative data was gathered through open-ended questionnaires and interview and analyzed textually to supplement the questionnaire survey.

To assess the distribution of the covariates between the two groups, the quantitative data underwent analysis using descriptive techniques. This involves examining the independent variables through methods such as calculating frequencies and percentages, which were then summarized in tables to interpret and draw conclusions from the results. Inferential statistical tools, including mean, Chi-square test, one-sample t-test, and paired t-test, were employed to determine the significance of mean values for continuous variables between the households of PW-PSNP beneficiaries and eligible non-beneficiaries. To estimate the impact of household's participation in the Urban Productive Safety Net program on asset accumulation and consumption expenditure logit model and an econometric model (Propensity score matching method) were used in addition to the simple descriptive analysis.

**2.4.1 Binary logistic regression model.** According to Gujarati [27], in estimating logit model for the dichotomous (binary) logistic regressions was used to achieve this objective, particularly; participation which takes a value of 1 if the household participated in a program and 0 otherwise. The logit model was selected for this study because the dependent variable is dichotomous in nature and the computation will be easier. For the binary treatment case, where we estimate the probability of participants versus non-participants of UPSNP is addressed as a decision involving binary/dichotomous response variable. The logit model is mathematically formulated as follow:

$$\text{Pi} = \frac{e^{Zi}}{1 + e^{zi}} \qquad (1)$$

Where, Pi -is the probability of participation in the PSNP in relation to explanatory variable $e^{zi}$ - irrational number to the power of Zi

$$Zi = \beta o + \Sigma\, Bi\, Xi + Ui \qquad (2)$$

Zi—a function of n explanatory variable 1, 2. . . . . . . . . . .n)
βo—is the intercept

Bi- is regression coefficients to be estimated

Xi- is pre-intervention characteristics (explanatory variable)

Ui- is a disturbance term

The probability that the household belongs to non-participant group is given by:

$$1-\text{Pi} = \frac{1}{1 + e^{zi}} \tag{3}$$

Then the odd ratio was written as:

$$\frac{\text{Pi}}{1 - \text{Pi}} = \frac{1 + e^{zi}}{1 + e}\text{-zi} = e^{zi} \tag{4}$$

$\frac{\text{Pi}}{1-\text{Pi}}$. It is simply the odds ratio in favor of participating in the UPSNP. It is the ratio of the probability that the household would participate in the UPSNP to the probability that he/she would not participate in the UPSNP.

By taking the natural log of Eq (4), the log of odds ratio was written as:

$$\text{Li} = \text{Ln}\left(\frac{\text{Pi}}{1 - \textbf{Pi}}\right) = \text{Ln}(e^{zi}) = \text{Zi} = \beta o + \Sigma\,\text{BiXi} + \text{Ui} \tag{5}$$

Li—is the log of the odds ratio in favor of participation in UPSNP which is linear both in Xi and parameters.

**2.4.2 Propensity Score Matching (PSM) method.** As placement to participate in PSNP is not random, i.e., an eligible household is deliberately selected based on asset size, vulnerability to shocks and food insecurity. In addition, the baseline survey was not conducted prior to the intervention of the PSNP in the study area. Therefore, propensity score matching method is usually used to assess the impact of a program, in this case, we used to address the impacts of PSNP on asset accumulation and consumption in the beneficiary households.

According to Gilligan et al. [28] this model assumes that, "after controlling for all pre-intervention observable household's and community's characteristics that are correlated with both program participation and the outcome variable, untreated groups have the same average outcome as the treated group would have had they did not receive the program". This is the basic idea, to found in group of non-participants those individuals who are similar to the participants in all relevant pre-treatment characteristics X. Hence, the matching approach is one of the best solutions.

*Procedures to estimate propensity score.* It is the first step of PSM technique. Rosenbaum and Rubin [29] revealed that matching can be performed conditioning only on P(X) rather than on X, where P(X) = Prob (D = 1|X) is the probability of participating in the program conditional on X. According to these authors, if outcomes without the intervention are independent of participation given X, then they are also independent of participation given P(X). Estimating the propensity score involves decision on three choices: what model to be used for the estimation and what variables should be included in this model and matching algorism.

*Impact indicator variables for asset accumulation and consumption.* The impact of the PSNP is analyzed by dividing the two categories which are the asset building and consumption expenditure of the PW-PSNP beneficiary and eligible non-beneficiary households [30].

*Outcome variables for asset accumulation.* The first is the home asset (which is the sum of household goods and consumer durables) and community asset value is also measured based on the estimation of its market value by respondents in terms of Birr [31].

*Outcome variables for consumption.* In order to analyze the impacts of the PSNP on the household's consumption the study has attempted to examine it's by classifying the

households' consumption into two main categories, because households have consumed different items and due to this reason their value might not the same. Therefore, in the study, the households have been asked about the quantity as well as the current price of overall households mean real food consumption expenditure per Adult Equivalent (AE) in terms of 24 hours and current non-food consumption expenditure per adult equivalent (AE) in terms of year, at their disposal, during the field survey based on the respondents' consumption rate in terms of Ethiopian Birr [32].

## 3. Results and discussions

### 3.1 Descriptive analysis of the dummy variables used in logit model

**Sex of the household head.**   In the urban area, female headed households are more effective in feeding their family in a sustainable manner from year to year which makes them more likely to be food secure. Because, female headed households are in a better labor input than the males headed in urban area. Based on the respondents result in this study, the sample was composed of both male and female headed households from the total sample households 94 (84%) and 18 (16%) were male and female headed household respectively. From the total male headed household respondents, that have been included in the study, 54 (48%) are from the PW-PSNP participant group while the other 40 (36%) are from the non-PSNP participants. In other way female headed households represent about 8 (7%) and 10 (10%) from PW-PSNP participant and non-PSNP beneficiary households respectively. As the result indicates that, in the study area the male headed households were a higher chance to PSNP participants in the public work components of the PSNP. Therefore, as compared to their female headed households' counterpart male headed households participate more in the PSNP.

**Credit experience.**   It is indicating whether the household had credit experience or not. Those who had credit experience can use it to purchase something so that they would manage to escape the liquidity constraints experienced by many urban households in the study area. Moreover, the credit could also serve as a capital source to fund any off-farm activity that the households may undertake indicating that credit experience minimizes the danger of being food unsecured. Amhara Credit and Saving Institution is one of the main and dominant credit suppliers in the study area. Prior to PSNP of the respondents only 11 (10%) PW-PSNP participant and 7 (6%) of non-PSNP participants had taken a loan. Prior to PSNP shows that about 51 (46%) of respondents and 43 (38%) of non-beneficiary respondents had not borrowing credit experience. Access to credit after the implementation of the program indicated that out of the total respondents 38 (34%) of them were public work of PSNP beneficiaries and 33 (29%) non-beneficiaries households received credit, mainly to purchase food item, and improve household asset. It indicates that, higher percent of PSNP public work participants had access to credit compared to non-participant groups. Hence, after the program, from the total respondents of public work PSNP beneficiary have been better and borrowing experience than non-beneficiary household. It indicates that greater percentage of beneficiary households mainly to purchase food item and improve home asset compared to non-beneficiary households. Moreover, the survey result revealed that out of the total respondents with off-farm income, 51 (46%) of them were from the participant group and the 41 (37%) were non-participant households. A study done by Derso et al. [33] supported this result. It indicated that three in four households were experiencing food insecurity and that an appropriate multi-dimensional approach such as the opening of small business enterprises through credit service support may help to diversify income sources and to reduce the dependency ratio. While, without off-farm income, 11 (10%) of them were from the participant group and the others 9 (8%) were found to be non-participant households; regarding access to extension service about 55

**Table 2. Descriptive analysis of the dummy variables used in the study.**

| Explanatory Variables | All samples | PSNP categories | | | |
|---|---|---|---|---|---|
| | | UPSNP beneficiary HHs | | Non-UPSNP beneficiary HHs | |
| | | Frequency | Percentage | Frequency | Percentage |
| Sex of household heads | Male (n = 94) | 54 | 48 | 40 | 35.7 |
| | Female (n = 18) | 8 | 7 | 10 | 9 |
| Access to credit before the implementation of the program | Yes (n = 18) | 11 | 10 | 7 | 10 |
| | No (n = 94) | 51 | 45.5 | 43 | 38 |
| Access to credit after the implementation of the program | Yes(n = 71) | 38 | 33.9 | 33 | 29.4 |
| | No(n = 41) | 24 | 21.4 | 17 | 15 |
| Off-farm income before program | Yes (n = 92) | 51 | 45.5 | 41 | 36.6 |
| | No (n = 20) | 11 | 10 | 9 | 8 |
| Access to extension services and training | Yes (n = 88) | 55 | 49 | 33 | 29.46 |
| | No (n = 24) | 7 | 6.25 | 17 | 15 |
| Household head education | Literate(n = 56) | 31 | 27.6 | 25 | 22.3 |
| | Illiterate(n = 56) | 31 | 27.6 | 25 | 22.3 |
| Access to remittance | Yes (n = 30) | 19 | 17 | 11 | 10 |
| | No (n = 82) | 43 | 38 | 39 | 35 |
| Experienced food gap | Yes (n = 106) | 58 | 52 | 48 | 43 |
| | No (n = 6) | 4 | 3.5 | 2 | 1.7 |

(49%) of PSNP in public work participants and 33 (29%) of non-participants were receiving the type of training in 2019/20. But, the remaining 7 (6%) of PSNP participants and 17 (15%) non-participants did not receive any type of training. Based on these results, to some extent the extension service was highly supportive to the PSNP beneficiary households when compared with non-beneficiary households. According to the survey result the educational status of the household revealed that 31 (27.6%) of PW-PSNP beneficiary households were literate (able to read and write) and 25 (22%) of non-beneficiary households were literate (able to read and write). While, 31 (27.6%) and 25 (22%) of PW-PSNP beneficiary and PSNP non-beneficiary household heads were illiterate (unable to read and write) respectively. The result indicated that, both illiterate and literate household heads in the respondent were similar in proportion (Table 2).

On the other hand, access to remittance the result of the study showed that about 51(45.5%) of the PW-PSNP beneficiary households did not receive any kind of remittance, and the rest of 11 (10%) received remittance mainly to purchase food item and none food items. while from the total none beneficiary households, 11 (10%) household heads who took remittances and the remaining 39 (35%) non-beneficiary household heads did not received remittance respectively. This means greater percentages of beneficiary of the UPSNP are not received remittance as compared to non-beneficiary. Besides, the result of the study also revealed that food gap faced households prior to the introduction of the program three month and above were faced food gap, 58 (52%) and 48 (43%) of PW-PSNP beneficiary and non-PSNP beneficiary households' groups had food gap on three months of experienced food gaps respectively. It implies that prior to the introduction of the program higher percentage of participant groups were faced food gap and food insecure compared to non-beneficiary households (Table 2 above).

### 3.2 Descriptive analysis of the continuous variables used in logit model

**Age of the household head.** It is a continuous variable that refers to the level of 'experience the farmers possess and measured in the number of years. The older the head of the

**Table 3. Descriptive analysis of continuous variables used in logit model.**

| Variables | All samples | Beneficiary | Non-beneficiary | Mean difference | T-value |
|---|---|---|---|---|---|
| Age of the HH head | 49.17 | 49.91 | 48.33 | 1.58 | 0.97 |
| Family size | 4.03 | 4.17 | 3.88 | 0.29 | 1.64 |
| Dependency ration | 1.25 | 1.40 | 1.08 | 0.32 | 2.63*** |
| Average family age | 30.28 | 30.52 | 30.02 | 0.49 | 0.41 |

Note

*** means significant at 1%

household is the more he/she would be effective in risk diversification as well as in predicting the weather and applying the right technology. However, the existence of large number of children under the age of 15 and old age of 65 and above in the family affect the household consumption negatively. The households with higher dependency ratio will have lowest welfare status. This means households with higher dependency ration had low level of welfare, hence higher probability to fall in to food in security and poverty. The result of the study indicated that the average age of the household head for the whole sample is around 49.17 years (Table 3). When we compare the two groups in terms of their mean age, it was found that treated groups are older than their controlled groups by about 1.58 year, i.e., 49.91 years compared to 48.33 years respectively. Thus, the older household heads were found to be food in secured.

**Dependency ration.** The dependency ratio shows the ratio of economically active persons compared to economically dependent household members. Economically active members of households, whose age is from 14 to 64 were assumed to be the principal productive force and sources of income for the household. Household members whose age was between 0–14 and above 64 were considered as economically inactive and dependent members of the household. Similarly, the dependency ratio for the members of the sampled households estimated to be 1.25 and 1.08 of the PW-UPSNP beneficiaries and non-UPSNP beneficiaries respectively. Whereas, the average difference between treated and non-treated group are around 0.32. Therefore, the result indicates that treated groups are the higher dependency than control groups. Thus, the higher dependency contributes to food insecurity and the probability level of participation significantly 1 percent (Table 3 above).

**Household family size.** It refers to the total number of the household's member. Family size is one of the major socioeconomic variables, which as a significant influence on the amount of household consumption. As the number of the family increases, given the fixed resources, the food share goes to the individual in that family declines which maximizes the probability of being food insecure. As presented in Table 3, the average family size of the combined samples were around 4. The average family size of beneficiary and non-beneficiary households was 4 and 3 respectively. As a result, beneficiary households had greater family size compared to non-beneficiary households (Table 3 above).

### 3.3 Impact of productive safety net program on community and home asset values

Different types of sustainable productive activities were being undertaken through public works component of PSNP to build community assets. These activities largely focused on the households' area of community asset like maintenance of roads, maintenance and construction of schools, maintenance and construction of latrine and cleaning surrounding environment. Whereas, home asset (consumer and durable assets like radio, wrist watch, bed, mobile

**Table 4. Mean comparisons for impact indicators under asset accumulation.**

| Name of the assets | All sample | UPSNP-beneficiary | Non-UPSNP beneficiary | Mean difference | T-value |
|---|---|---|---|---|---|
| Current home asset value | 496.05 | 711.63 | 254.59 | 457.04 | 6.99*** |
| Community asset value | 319.95 | 345.66 | 291.15 | 54.51 | 2.64*** |

Note

*** means significant at 1% level of significant

phone, Tape recorder, Television and housing condition) were included. Then, the result of the study revealed that in the households' area of community asset and home asset value calculated as the beneficiary worked and contribute as estimated at the values have 345.6 and 711.6 Ethiopian Birr respectively. Whereas, non-beneficiary households of community asset and home asset value contribute to have 291.1 and 254.6 Ethiopian Birr respectively. Therefore, the beneficiary households had greater amount of community and home asset building tendency than non-beneficiary households (Table 4). According to this finding, the mean difference in community asset and home asset value contribute between the beneficiary and non-beneficiary households were found to be positive and statistically significant at 1% of probability level. This implies that, the PSNP beneficiary households possess community asset and home asset value that have contribute to greater value (quality) as compared to the assets possessed by the non-PSNP beneficiary households.

## 3.4 Impact of productive safety net program on consumption

Consumption expenditure was used as impact indicator while evaluating impact of the UPSNP, and it was computed as per adult equivalent consumption expenditure. The overall households mean real consumption expenditure per adult equivalent (AE) for the sample households was 257.63 Ethiopian birr. The mean consumption expenditure for PSNP beneficiaries and non-beneficiary groups was 295.21 and 215.54 Ethiopian Birr respectively. The mean difference test of consumption expenditure for the two groups was 79.68 Ethiopian Birr. It implies that statistically significant at 1% probability level. Whereas the overall households mean real food consumption and non-food consumption expenditure per adult equivalent (AE) for the sample households was 209.95 and 47.68 Ethiopian Birr respectively (Table 5). The mean food consumption expenditure for PSNP beneficiary and non-beneficiary groups was also 237.72 and 178.86 Ethiopian Birr respectively. The mean difference test of consumption expenditure for the two groups was 58.86 Ethiopian Birr. It implies that statistically significant at 1% probability level. Moreover, the mean non-food consumption expenditure for PSNP beneficiary and non-beneficiary groups were 57.49 and 36.68 Ethiopian Birr respectively. Therefore, the result of the study showed that the PSNP beneficiary households were found to be positive and statistically significant at 1% over the intervention period. Besides, the

**Table 5. Mean comparisons on the impact of UPSNP on consumptions per adult equivalent household.**

| Variables | All samples | Public work beneficiary | Non-PSNP beneficiary | Mean difference | T-value |
|---|---|---|---|---|---|
| Consumption expenditure per adult equivalent household | 257.63 | 295.21 | 215.54 | 79.68 | 3.88*** |
| Food consumption expenditure per adult equivalent household | 209.95 | 237.72 | 178.86 | 58.86 | 3.04*** |
| Non- food consumption expenditure per adult equivalent household | 47.68 | 57.49 | 36.68 | 20.81 | 5.83*** |

Note

*** means significant at 1% level of significant

overall consumption per adult equivalent for the participants' increased when compared to non-participants over the intervention period. This result was supported by the evidences that taken from the key informants at the time of interview. One of the key informants said that:

> *"The beneficiary households have better money to pay for food items, medical cost and school fee with the help of PW-UPSNP. Before they included in PSNP, some of the beneficiary households were eating once a day but some others were eating twice in a day. After they were included in the PW-UPSNP, the majority of the beneficiary households are eating three times in a day".*

## 3.5 Econometric results

The study used propensity score matching to evaluate the effects of PSNP on asset accumulation and consumption expenditure among urban households. The propensity score was estimated using a logistic model based on observable characteristics to determine the likelihood of households participating in the program. Beneficiary and non-beneficiary households were then matched based on their propensity scores, using a binary outcome variable (1 for PW-PSNP beneficiary households and 0 for eligible non-PSNP beneficiary households) in the logit model.

**3.5.1 Factors of participation in productive safety net program.** The researchers have incorporated relevant variables to explain the dependent variable and conducted a logit model. They found that the gender, age of the household head, family size, and off-farm income activities significantly affected program placement during selection.

The findings show that various factors significantly affected participation in PW-PSNP. Employment on wage, family size, number of adult females, age of household head, and employment on own business before UPSNP intervention all had positive and significant influences on PW-PSNP participation, with varying magnitudes of impact.

Households with larger family sizes are more likely to participate in the program compared to those with smaller family sizes. This is because larger families often have higher food demand and a higher chance of food insecurity due to limited resources. The study found family size to be statistically significant at a 5% level of significance. This result is consistent with the studies conducted in Kenya and other rural parts of north Ethiopia by [34, 35] found that households with a large family size were more like to be food secure as compared to those with a small family size. The reason may be as the household size becomes larger, the number of economically active members may also increase, which leads to increase their participation in various income-generating activities that could boost their food security than for households with lower family size. Additionally, male-headed households were more likely to be program participants compared to female-headed households, likely due to females being active in non-farm activities generating off-farm income. This result was also statistically significant at a 5% level of significance.

Household head age is the other variable which is found to be negatively significant at 1% level of significant and related with probability of participation in PW-PSNP. This implies that as increase in age of household heads, decreases their participation in the program. This is due to the reason that, the age goes beyond certain periods; the person may become physically weak and food insecure. On the other hand, variables like sex of household head, food gap, accesses to extension services and access to remittance and number of adult males are found to be statistically insignificant (Table 6).

**3.5.2 Impact of UPSNP on household asset accumulation.** In this study, household home assets represent the sum of household goods like home, pot, oven, bed, box and barrel

**Table 6. Logit model estimates for household's participation in UPSNP.**

| Explanatory variables | Coefficient | Roust Std. Err. | P- Value |
|---|---|---|---|
| Sex of household head | -.647086 | .4778888 | 0.176 |
| Age of household head | -.3801537 | .1072199 | 0.000*** |
| Food gap before the implementation of the program | 1.521904 | 1.177113 | 0.196 |
| Employment on wage before the implementation of the program | .8004876 | .4212125 | 0.057* |
| Employment on own business before the implementation of the program | 1.241627 | .456983 | 0.007* |
| Accesses to extension services | .4487181 | .3334611 | 0.178 |
| Access to credit before | -.2792303 | .4296875 | 0.516 |
| Access to remittance the implementation of the program | -.3621517 | .4033549 | 0.369 |
| Education of household head | .3446515 | .3264757 | 0.291 |
| Number of adult male | -.1649666 | .2454619 | 0.502 |
| Number of adult female | -.5768482 | .2447218 | 0.018** |
| Family size | .4680607 | .1911481 | 0.014** |
| Constant | 6.996062 | 3.094728 | 0.024 |

Note

*, **, *** means significant at 10%, 5% and 1% respectively

and consumer durables like mobile phone, watch, tape recorder and Radio and Television etc. After converting both the household goods and the consumer durable of the entire households in to their respective market value, the two categories have been added together to form the household home assets.

The impact of UPSNP on household's home asset were found to be positive with the maximum and the minimum ATT value calculated to be 494 Ethiopian Birr (based on NN matching) and 441 Ethiopian Birr (based on Kernel matching) (Table 7) respectively. Statistically, this result is found to be significant at one percent for all matching estimators used in this study, i.e., based on Kernel, Stratification and NN matching. The implication of this result is, because of the introduction of the program, the value of non-productive/home asset at the hands of the PW-PSNP beneficiary households is higher than the value of the non-productive asset at the hands of the non-beneficiary households by the amount of at maximum 494 Ethiopian Birr and at minimum 441 Ethiopian Birr. And this result confirms the fact that the program is successful in avoiding the depletion in the non-productive asset of the beneficiary households.

According to this finding, the program has a positive and significant impact on the household asset at 1% level of significance. This can be also stated as PW-PSNP beneficiary

**Table 7. ATT estimated results of the impact of UPSNP on asset accumulation.**

| Outcome variables | Matching estimators | PW UPSNP beneficiary | Eligible non-PSNP beneficiary | Average treatment effect on the treated (ATT) | T-statistics |
|---|---|---|---|---|---|
| Current value of home asset | Nearest neighbor | 62 | 23 | 494.40 | 6.47*** |
| | Stratification | 62 | 49 | 455.37 | 7.24*** |
| | Kernel | 62 | 49 | 441.57 | 6.01*** |
| Community asset | Nearest neighbor | 62 | 23 | 82.88 | 2.13** |
| | Stratification | 62 | 49 | 54.44 | 2.61*** |
| | Kernel | 62 | 49 | 59.72 | 2.49*** |

Note

**, *** means significant at 5% and 1% respectively

households have more asset value than the eligible non-beneficiary households. Generally, the above positive value of ATT implies that, because of the program intervention household's home assets have been protected in the study area.

During the field survey, the participants of the PW-PSNP program made efforts to enhance community assets, as evidenced by their contributions to activities such as road maintenance, improving the greenery of the environment, and erecting fences around schools, health posts, and ponds. The result of this study shows that the current value of community assets at the hands of the PSNP beneficiary households is higher than the current value of community assets at the hand of the non-PSNP beneficiary households.

The mean difference in the current value of community assets between the PW-PSNP beneficiary and eligible non-beneficiary households were found to be positive and significant with the minimum and maximum ATT value of contribute as calculated 54 Ethiopian Birr (based on Stratification matching) and 82 Ethiopian Birr (based on the NN matching) respectively. Statistically, the calculated ATT for the treated and control group is found to be significant at 1% based on Kernel and Stratification matching while 5% with nearest neighbor matching estimators used in the study.

According to the result of this study on average the community assets of the PW-PSNP beneficiary households contribute to the value at maximum 82 Ethiopian Birr (based on NN matching) and to the minimum 54 Ethiopian Birr (based on Stratification matching) more as compared to the community assets of the eligible non- beneficiary households.

In general, because of their participation in the program the PW-PSNP beneficiary households have managed to protect their community assets from possible deterioration. Thus, the program enables the PSNP beneficiary households to preserve their community assets from depletion. The result coincides with [28], who assessed the impact of Ethiopia's PSNP using the method of propensity score matching method based on various impact indicators, also found that the mean difference in community assets between PW-PSNP and eligible non-PSNP households are positive and significant at one percent. Therefore, the result of the study showed that participation in PW-PSNP was found to be helpful to the participants in improving their community assets. This result is also coinciding with the finding of Yibrah [36], who studied the impact of participating in the Productive Safety Net Program on Consumption and Asset holding.

**3.5.3 Food security status of sample households.** Data on food consumption was collected from 112 households for this study, aiming to capture the diversity and frequency of various foods consumed within a 24-hour period. The study utilized a household food consumption score, which is a continuous measure and can be divided into three levels. The results indicated that 45 households (72.5%) of PW-UPSNP beneficiaries had acceptable food consumption, 11 households (17.7%) had borderline consumption, and 6 households (9.6%) were considered poor in terms of their food consumption score. On the one hand, among eligible non-UPSNP beneficiary households, 38% had acceptable food consumption, 36% had borderline consumption, and 26% had poor consumption. The households with poor consumption are considered food insecure, while those with borderline consumption are moderately food insecure, and the households with adequate food consumption are food secure. On the other hand, the mean food consumption score for PW-UPSNP beneficiary households was 1.53, whereas for eligible non-PSNP beneficiary households, it was 1.23. Consequently, there is a significant difference in the mean food consumption score between PW-UPSNP beneficiary households and eligible non-UPSNP beneficiary households, which amounts to 1.38.

The outcome suggests that households benefiting from PW-UPSNP are more food secure compared to those eligible for PSNP but not receiving PW-UPSNP. The chi-square test

**Table 8. Household food security status.**

| Food consumption score | PWUPSNP of beneficiary | | Eligible-non PWUPSNP beneficiary | | Total | | X²-value |
|---|---|---|---|---|---|---|---|
| | Frequency | % | Frequency | % | Frequency | % | |
| Acceptable food consumption (>42) | 45 | 72.5 | 19 | 38 | 64 | 57 | |
| Borderline food consumption (28.5–42) | 11 | 17.7 | 18 | 36 | 29 | 26 | |
| Poor food consumption (≤28) | 6 | 9.6 | 13 | 26 | 19 | 17 | |
| Total | 62 | 100 | 50 | 100 | 112 | 100 | |
| Mean | 1.53 | | 1.23 | | 1.38 | | 15.06 *** |

Note

*** Significant at 1% level

confirms a significant difference in food security status between the two groups at a 1% level of significance (Table 8).

**3.5.4 Food consumption expenditure per adult equivalent.** Food consumption expenditure behavior can be used to assess household food security. The average food consumption expenditure for households benefiting from PW-UPSNP was 237.72 Ethiopian Birr, while for non-beneficiary households, it was 178.86 Ethiopian Birr. The statistically significant difference of 58.86 Ethiopian Birr between the two groups was observed at a 1% probability level (Table 9).

Moreover, this result indicates that the PW-UPSNP on beneficiary households food consumption had a positive with the maximum and the minimum ATT value calculated to be 59 Ethiopian Birr (based on stratification matching) and 45 Ethiopian Birr (based on NN matching) respectively (Table 10).

# 4. Conclusions

Chronic poverty and lack of food are major global problems, particularly in developing regions. In Ethiopia, around 8.5 million people, constituting 10% of the population, suffer from chronic food insecurity, leading to asset selling and consumption restrictions. A study shows that factors like household head's age, family size, and off-farm income activities influence participation in the PW-UPSNP program. Beneficiary households have more assets than eligible non-beneficiary households. The PW-UPSNP program significantly increases food consumption, non-food consumption, and total consumption expenditure for beneficiaries compared to non-beneficiaries.

Econometric analysis reveals positive effects of the PW-UPSNP program on community and home asset accumulation. More PW-UPSNP beneficiaries are food secure compared to non-beneficiaries. The mean food consumption expenditure is significantly higher for PW-UPSNP beneficiaries. The program has a positive and statistically significant impact on

**Table 9. Mean Comparisons on the impact of PSNP on consumptions per adult equivalent households.**

| Variables | All sample mean | PWUPSNP beneficiary | eligible-non-UPSNP beneficiary | Mean difference | T-value |
|---|---|---|---|---|---|
| Food consumption expenditure per adult equivalent household | 209.95 | 237.72 | 178.86 | 58.86 | 3.04*** |

Note

*** means significant at 1% level of significant

**Table 10. ATT estimated result of the impact of PSNP on household consumption expenditure.**

| Outcome variables | Matching estimators | PW UPSNP beneficiary | eligible non-UPSNP beneficiary | Average treatment effect on treated | t-statistics |
|---|---|---|---|---|---|
| Food consumption expenditure per adult equivalence | Nearest neighbor | 62 | 23 | 45.6 | 1.51 |
| | Stratification | 62 | 49 | 59.63 | 3.19*** |
| | Kernel | 62 | 49 | 54.07 | 2.71*** |

Note

*** Significant at 1% level

food consumption. The second phase of the PSNP program in Ethiopia ensures food security for both chronically and temporarily food insecure households. It aims to assist over 7.5 million people across eight regions through program transfers and productive activities.

Previous emergency food relief efforts were insufficient, leading to the implementation of the PSNP program. It is the largest social protection program in Sub-Saharan Africa and focuses on selected households over several years to achieve sustainable food security. Most studies have focused on rural areas, but the overall conclusion is that the PW-UPSNP program has a positive effect on consumption, asset accumulation, and food security. The program successfully addresses chronic poverty and food insecurity in Ethiopia.

Future studies could examine the long-term impacts of the urban productive safety net program, including its effects on household resilience, poverty reduction, and broader community-level outcomes over time. Researchers could also investigate potential barriers to program participation and ways to improve access and targeting for the most vulnerable urban households. Comparative analyses with rural productive safety net programs could provide insights into contextual factors influencing program effectiveness in urban versus rural settings. Additionally, qualitative research exploring participants' experiences and perceptions could complement the quantitative findings to offer a more holistic understanding of the program's impacts.

## Acknowledgments

The author would like to thank farmers, agricultural development agents, and local administrators of the study area for their assistance during the field work.

## Author Contributions

**Supervision:** Alem-meta Assefa.

**Writing – review & editing:** Alem-meta Assefa.

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
