## [Decision Letter · Decision Letter 0]

14 Feb 2024

PONE-D-23-24451Impact of Urban Productive Safety Net Program on Urban Households’ Asset Accumulation and Food Consumption Rate in Dessie City, South Wollo Zone, Amhara Region, EthiopiaPLOS ONE

Dear Dr. Assefa,

Thank you for submitting your manuscript to PLOS ONE. After careful consideration, we feel that it has merit but does not fully meet PLOS ONE’s publication criteria as it currently stands. Therefore, we invite you to submit a revised version of the manuscript that addresses the points raised during the review process.

We look forward to receiving your revised manuscript.

Kind regards,

Olutosin Ademola Otekunrin

Academic Editor

PLOS ONE

Journal Requirements:

4. In your Methods section, please provide additional information regarding the permits you obtained for the work. Please ensure you have included the full name of the authority that approved the field site access and, if no permits were required, a brief statement explaining why.

5. You indicated that ethical approval was not necessary for your study. We understand that the framework for ethical oversight requirements for studies of this type may differ depending on the setting and we would appreciate some further clarification regarding your research. Could you please provide further details on why your study is exempt from the need for approval and confirmation from your institutional review board or research ethics committee (e.g., in the form of a letter or email correspondence) that ethics review was not necessary for this study? Please include a copy of the correspondence as an ""Other"" file.

6. Thank you for including your ethics statement:  "N/A".   

1. For studies reporting research involving human participants, PLOS ONE requires authors to confirm that this specific study was reviewed and approved by an institutional review board (ethics committee) before the study began. Please provide the specific name of the ethics committee/IRB that approved your study, or explain why you did not seek approval in this case.

7. In this instance it seems there may be acceptable restrictions in place that prevent the public sharing of your minimal data. However, in line with our goal of ensuring long-term data availability to all interested researchers, PLOS’ Data Policy states that authors cannot be the sole named individuals responsible for ensuring data access (http://journals.plos.org/plosone/s/data-availability#loc-acceptable-data-sharing-methods).

8. We note that Figure 2 in your submission contain map/satellite images which may be copyrighted. All PLOS content is published under the Creative Commons Attribution License (CC BY 4.0), which means that the manuscript, images, and Supporting Information files will be freely available online, and any third party is permitted to access, download, copy, distribute, and use these materials in any way, even commercially, with proper attribution. For these reasons, we cannot publish previously copyrighted maps or satellite images created using proprietary data, such as Google software (Google Maps, Street View, and Earth). For more information, see our copyright guidelines: http://journals.plos.org/plosone/s/licenses-and-copyright.

9. Please include your tables as part of your main manuscript and remove the individual files. Please note that supplementary tables (should remain/ be uploaded) as separate "supporting information" files

Additional Editor Comments:

 l am pleased to inform you that three anonymous reviewers have reviewed your manuscript and you are expected to attend to their comments and suggestions as early as you can.

Reviewers' comments:

Reviewer's Responses to Questions

Comments to the Author

1. Is the manuscript technically sound, and do the data support the conclusions?

Reviewer #1: Partly

Reviewer #2: Yes

Reviewer #3: Yes

2. Has the statistical analysis been performed appropriately and rigorously? 

Reviewer #1: No

Reviewer #2: Yes

Reviewer #3: Yes

3. Have the authors made all data underlying the findings in their manuscript fully available?

Reviewer #1: Yes

Reviewer #2: Yes

Reviewer #3: Yes

4. Is the manuscript presented in an intelligible fashion and written in standard English?

Reviewer #1: Yes

Reviewer #2: Yes

Reviewer #3: Yes

5. Review Comments to the Author

Reviewer #1: The paper is interesting. The results could be used to justify the continuity of a program to help poor households, or to encourage the scaling up of such a program.

Abstract

Specify the number of beneficiary and non-beneficiary surveyed households

Specify econometric method

Introduction

The background is well-developed.

Strengthen the literature review.

Material and method

The program targets 11 cities, better justify the choice of Dessie.

Better justify the choice of Dawudo as a sub-city.

In Dawudo, specify how many districts there are (including robit Kebele and Bahil Amba Kebele).

What are the control variables? Justify the choice of these variables

The logit method is not clear. Why do you study the determinants of program participation? It is an impact study.

Perform robustness or sensitivity tests

Results

Some information in the results should be included in the methodology, such as the description of variables and expected impacts.

It is said that “The study used mixed research approach” (Line 142), but there is no qualitative analysis in the results (thematic analysis or verbatim, etc.).

To ensure the veracity of the results, include the results of a sensitivity test.

Conclusion

Reinforce the discussion

Clarify research limitations

Compare results with those of other authors

Do not repeat results

Reviewer #2: •In the abstract section, rather than saying ‘’demographic and socio-economic variables such as’’ it is better to mentioning the significant variable. Furthermore, incorporate the recommendation in the abstract section in short.

•Write the structure of the study in introduction section .How many Safety Net Program users in Amhara region?

•Write the description of and measurement of variables used in the study in the methodology section.

•It better to write hypothesis explanation and expected sign of the variables in the methodology section.

•Support your finding by empirical literature.

•It is better to write conclusion and recommendation/ policy implication separately.

•Add limitation of the study and future study area in the conclusion and policy implication section.

Reviewer #3: REVIEWER’S COMMENT

Summary

Chronic poverty, particularly the long-term lack of food, remains a significant issue. Millions of people worldwide faced food insecurity, with a majority residing in developing regions. Ethiopia, in particular, had about 8.5 million chronically food insecure individuals, constituting 10 percent of its population. These individuals lacked consistent access to sufficient food, leading to asset selling and consumption restrictions in order to survive. A study conducted indicated that the second phase of Productive Safety Net Programmed (PSNP) plays a crucial role in ensuring food security for both chronically and temporarily food insecure households in Ethiopia. This program aims to assist over 7.5 million people (1.6 million households) in 318 districts across eight regions by providing program transfers and supporting productive activities until December 2014. Before implementing PSNP, the Ethiopian government relied on unpredictable annual appeals for emergency food relief, which saved lives but failed to address the increasing number of food insecure individuals. The implementation of PSNP was done in an attempt to break the cycle of annual appeal of food aid and achieve an acceptable level of food security at macro (national) and micro (household) level. The PSNP is currently the largest operating social protection program in Sub-Saharan Africa, outside of South Africa. It differs from previous food-for-work programs, in that it focuses continuously on selected household over several years and in that the explicit objective was that it will eventually be phased out. At national level, there were some works that tried to highlights the impacts of program intervention, particularly PSNP. Unfortunately, most of them have focused on the impacts of PSNP on asset building and consumption in rural areas rather than urban areas. Hence, this study aimed at bridging the knowledge gap.

Major comments

The introduction, methods, and results, sections are well documented with good clarity and are scientific facts. However, there are major revisions to be done on the manuscript before publication.

Minor comments

1.In the abstract, the authors should remove italics from this section.

2.In the introduction, line 53, the name of the author should be mention before the reference number.

3.Lines 59 &60, approximately 850 million people worldwide faced food insecurity, with a majority residing in developing regions.

4.Lines 60 &61, In Africa, around 200 million people, accounting for 27.4 percent of the population, experienced food insecurity between the 1970s and 1990s

5.Lines 61 & 62, Ethiopia, in particular, had about 8.5 million chronically food insecure individuals, constituting 10 percent of its population.

6.Lines 65 &66 conducted a study indicating that the second phase of PSNP plays a crucial role in ensuring food security for both chronically and temporarily food insecure households in Ethiopia.

7. Lines 67 &68 districts across eight regions by providing program .transfers and supporting productive activities until December 2014.

8.Line 82, million urban poor living in 972 cities and towns.

9.Lines 86-88, beneficiaries, In the first phase, 604,000 the poorest 12 percent and about 55 percent of people living below the poverty line in these 11 cities Urban Productive Safety Net Project [10].

10.Line 93, Having these in mind the researcher was try to evaluate the impacts of……………………

11.Line 98 Chronically, food insecurity has been one of the most problems of urban poverty, particularly in drought-prone

12.Line 100, High proportion of food insecure households, which have been receiving emergency food aids.

13.Line 101, Although the emergency assistance was substantial and saved many lives, it was uncertainty of the food aid,

14.Lines 107-109, aim of preventing asset depletion at household level while creating assets at the community level (NO REFERENCE).

15.Lines 112-113, Some findings state that, PSNP has able to smooth the consumption and protects household’s assets (GRAMMER CHECK for clarity).

16.Line 113, [14] indicated that beneficiary households. Mention the authors NAME BEFORE the reference number.

17.Lines 114-115, In contrast to these, what proved is that there is no significant difference between, the consumption as well as assets of PSNP participants and non-participants.

18.Lines 116-118, Moreover, [17], stated that, participation in PSNP reduces the asset size of the participants, due to the dependency behavior of beneficiaries which discourage them to put efforts on their agricultural activities.

19.Line 112, levels have brought inconsistent results and far from conclusive,(This statement is not clear).

20.Line 136, THE RESEARCH DESIGN AND SAMPLING

21.Lines 137-139, Because of survey design helped identify patterns of behavior, activities, people‘s attitude, opinions, habits and any of the variety of social problems that influence the explicability and scaling-up efforts of the program across various contexts

22.Lines 139-141, Descriptive survey design help to identify the issues had been occurred and could be predetermine certain preventive measures could be adopted.

23.Line 144, The quantitative data were analyzed and interpreted in the numerical data through appropriate.

24.Line 154, stated that target population as all members of a real set of subjects to which a researcher wishes to…….

25.Lines 155-156, The study was targeted each selected Kebele, the name of the entire list of household heads were obtained from Kebele administration office.

26.Line 157-158, Then on the bases of Kebele document there are 338 PWPSNP beneficiary households and 272 eligible non-beneficiary households in Robit Kebele.

27.Line 160, beneficiary household heads were used to as sample frames included for the survey (The statement needs to be improved upon grammatically).

28.Line 175, In addition to this, 6 key informant interviewees were conducted with purposively.

29.Lines 175-176, ). In addition to this, 6 key informant interviewees were conducted with purposively selected Kebele Development Agents, Kebele Administrators and head of UPSNP Task Force.

30.Line 178, The study was employed both primary and secondary data sources.

31.Line184, The study was employed both quantitative and qualitative data analysis techniques

32.Lines 186 &187, Similarly, qualitative data was gathered through open-ended questionnaires and interview and analyzed through textually to supplement the questionnaire survey

33.Lines 189 &190, please rephrase

34.Line 190, and percentages were done to …..

35.Lines 193 &194, In addition, to the simple descriptive analysis, to estimate the impact of participation in the urban Productive Safety Net program.

36.Line 198, According to [22], in estimating logit model for the dichotomous (binary) logistic regressions will be used to

37.Line 249, by classifying the households’ consumption in to two main

38.Line 254, the field survey based on the respondents’ their consumption rate in terms of Ethiopian birr.

39.Results and Discussion section (pages 10- 17) should be well written with appropriate comparison with relevant previous research output/published articles and must be well referenced. That is this section should not just be result presentations alone without being adequately discussed.

40.Line 474, CONCLUSION

41.Line 534, (pages 20-21) REFERENCES

42.Conclusion/recommendation

Generally, the study titled “Impact of Urban Productive Safety Net Program on Urban Households’ Asset Accumulation and Food Consumption Rate in Dessie City, South Wollo Zone, Amhara Region, Ethiopia” was well-researched, very relevant, scientific, and very informative, it raised pertinent issues regarding the impact of government intervention on asset acquisition and food consumption in achieving the “zero hunger” which is the Sustainable Development Goal 2. The authors demonstrated excellent mastery of the study. However, major corrections are required as highlighted.

6. PLOS authors have the option to publish the peer review history of their article (what does this mean?). If published, this will include your full peer review and any attached files.

Do you want your identity to be public for this peer review? For information about this choice, including consent withdrawal, please see our Privacy Policy.

Reviewer #1: No

Reviewer #2: No

Reviewer #3: No

---

## [Author Response · Author response to Decision Letter 0]

29 Mar 2024

Alem-meta Assefa

Department of Geography and Environmental Studies, Wollo University

Dessie, Ethiopia

5 March 2024

TO: PLOS ONE

Re: Rebuttal Letter for the article titled "Impact of Urban Productive Safety Net Program on Urban Households’ Asset Accumulation and Food Consumption Rate in Dessie City, South Wollo Zone, Amhara Region, Ethiopia"

Dear Editor and Reviewers,

We would like to express our sincere gratitude to the academic editor and reviewer(s) for their valuable time and insightful comments on our manuscript titled "Impact of Urban Productive Safety Net Program on Urban Households’ Asset Accumulation and Food Consumption Rate in Dessie City, South Wollo Zone, Amhara Region, Ethiopia." We appreciate the effort they have put into reviewing our work and providing constructive feedback. In response, we have carefully addressed each point raised and made the necessary revisions to improve the quality of our manuscript.

In conclusion, we would like to express our gratitude to the academic editor and reviewer(s) once again for their invaluable feedback. We believe that the revisions we have made in response to their comments have significantly enhanced the quality and clarity of our manuscript. We are confident that the revised version of our article makes a valuable contribution to the field and is suitable for publication in PLOS ONE.

Thank you for considering our rebuttal and revised manuscript. We look forward to hearing from you regarding the final decision on our submission.

Sincerely,

Alem-meta Assefa

Department of Geography and Environmental Studies, Wollo University, Dessie, Ethiopia

Corresponding Email Address: alemmeta99@yahoo.com

---

## [Decision Letter · Decision Letter 1]

3 Jun 2024

PONE-D-23-24451R1Impact of Urban Productive Safety Net Program on Urban Households’ Asset Accumulation and Food Consumption Rate in Dessie City, South Wollo Zone, Amhara Region, EthiopiaPLOS ONE

Dear Dr. Assefa,

Thank you for submitting your manuscript to PLOS ONE. After careful consideration, we feel that it has merit but does not fully meet PLOS ONE’s publication criteria as it currently stands. Therefore, we invite you to submit a revised version of the manuscript that addresses the points raised during the review process.

**ACADEMIC EDITOR: I am pleased to inform you that experts in the field have reviewed your manuscript and you are expected to address their comments as early as possible. Thank you./>==============================**

**Please submit your revised manuscript by **Jul 18 2024 11:59PM**. If you will need more time than this to complete your revisions, please reply to this message or contact the journal office at plosone@plos.org. When you're ready to submit your revision, log on to https://www.editorialmanager.com/pone/ and select the 'Submissions Needing Revision' folder to locate your manuscript file**.

**Please include the following items when submitting your revised manuscript:**

**A rebuttal letter that responds to each point raised by the academic editor and reviewer(s). You should upload this letter as a separate file labeled 'Response to Reviewers'.**

**A marked-up copy of your manuscript that highlights changes made to the original version. You should upload this as a separate file labeled 'Revised Manuscript with Track Changes'.**

**An unmarked version of your revised paper without tracked changes. You should upload this as a separate file labeled 'Manuscript'.**

****

**If applicable, we recommend that you deposit your laboratory protocols in protocols.io to enhance the reproducibility of your results. Protocols.io assigns your protocol its own identifier (DOI) so that it can be cited independently in the future. For instructions see: https://journals.plos.org/plosone/s/submission-guidelines#loc-laboratory-protocols. Additionally, PLOS ONE offers an option for publishing peer-reviewed Lab Protocol articles, which describe protocols hosted on protocols.io. Read more information on sharing protocols at https://plos.org/protocols?utm_medium=editorial-email&utm_source=authorletters&utm_campaign=protocols**.

**We look forward to receiving your revised manuscript.**

**Kind regards,**

**Olutosin Ademola Otekunrin**

**Academic Editor**

**PLOS ONE**

**Journal Requirements**:

**Please review your reference list to ensure that it is complete and correct. If you have cited papers that have been retracted, please include the rationale for doing so in the manuscript text, or remove these references and replace them with relevant current references. Any changes to the reference list should be mentioned in the rebuttal letter that accompanies your revised manuscript. If you need to cite a retracted article, indicate the article’s retracted status in the References list and also include a citation and full reference for the retraction notice**.

****

**Reviewers' comments**:

**Reviewer's Responses to Questions**

**Comments to the Author**

**1. If the authors have adequately addressed your comments raised in a previous round of review and you feel that this manuscript is now acceptable for publication, you may indicate that here to bypass the “Comments to the Author” section, enter your conflict of interest statement in the “Confidential to Editor” section, and submit your "Accept" recommendation.**

**Reviewer #1: All comments have been addressed**

**Reviewer #3: (No Response)**

**2. Is the manuscript technically sound, and do the data support the conclusions?**

**The manuscript must describe a technically sound piece of scientific research with data that supports the conclusions. Experiments must have been conducted rigorously, with appropriate controls, replication, and sample sizes. The conclusions must be drawn appropriately based on the data presented. **

**Reviewer #1: Yes**

**Reviewer #3: Yes**

**3. Has the statistical analysis been performed appropriately and rigorously? **

**Reviewer #1: Yes**

**Reviewer #3: Yes**

**4. Have the authors made all data underlying the findings in their manuscript fully available?**

**The PLOS Data policy requires authors to make all data underlying the findings described in their manuscript fully available without restriction, with rare exception (please refer to the Data Availability Statement in the manuscript PDF file). The data should be provided as part of the manuscript or its supporting information, or deposited to a public repository. For example, in addition to summary statistics, the data points behind means, medians and variance measures should be available. If there are restrictions on publicly sharing data—e.g. participant privacy or use of data from a third party—those must be specified.**

**Reviewer #1: Yes**

**Reviewer #3: Yes**

**5. Is the manuscript presented in an intelligible fashion and written in standard English?**

**PLOS ONE does not copyedit accepted manuscripts, so the language in submitted articles must be clear, correct, and unambiguous. Any typographical or grammatical errors should be corrected at revision, so please note any specific errors here.**

**Reviewer #1: Yes**

**Reviewer #3: Yes**

**6. Review Comments to the Author**

**Please use the space provided to explain your answers to the questions above. You may also include additional comments for the author, including concerns about dual publication, research ethics, or publication ethics. (Please upload your review as an attachment if it exceeds 20,000 characters)**

**Reviewer #1: The results must be discussed in greater depth: are the results consistent with the results of which studies? What are the limitations of the work?**

**Reviewer #3: REVIEWER’S COMMENT**

Generally, the study titled “Impact of Urban Productive Safety Net Program on Urban Households’ Asset Accumulation and Food Consumption Rate in Dessie City, South Wollo Zone, Amhara Region, Ethiopia” was well-researched, very relevant, scientific, and very informative. The study focused how to achieve the “zero hunger” which is the Sustainable Development Goal 2 in Ethiopia and other sub-Saharan African Countries where food insecurity and hunger is thriving due to poor food production and agricultural practices.

The authors demonstrated excellent mastery of the study. However, few minor corrections are needed before the manuscript to proceeds to the production phase.

Please find below the area that needs clarification and improvement.

In the abstract, the authors should do some grammar checks on some tenses (appropriate use of past tense). And indicate the knowledge gap and/ problem statement.

Lines 67 &68 An estimated 149 million Africans are facing acute food insecurity—an increase of 12 million people from a year ago.

Lines 78 & 79 Before implementing PSNP, the Ethiopian 69 government relied on unpredictable annual appeals for emergency food relief, which saved lives but failed to 70 address the increasing number of…………………………………

Lines 84 to 89 The study suggests that the PSNP plays a crucial role in ensuring food

security for both chronically and temporarily food insecure households. By providing program

transfers and supporting productive activities, the PSNP aims to assist over 7.5 million people in 318 districts across eight regions. This targeted approach is likely more effective in addressing the needs of food insecure individuals compared to the unpredictable annual appeals for emergency food relief.

Lines 90 to 93 Unlike the initial emergency relief appeal method, which relied on unpredictable annual appeals for food aid, the PSNP operates on a set timeline until December 2014. This longer-term planning allows for more stability and predictability in addressing food insecurity issues.

Lines 126 &127 ]. Hence, Dessie City has been labeled as typical food insecure area despite various food security interventions with PSNP is effective or not……….

Lines 134 &135 Historically, urban poverty has consistently faced the significant challenge of food insecurity,

especially in drought-prone areas of Ethiopia

Methodology

Lines 296-302 Impact Indicator Variables for Asset Accumulation and consumption

The impact of the PSNP is analyzed by dividing the two categories which are the asset building

and consumption expenditure of the PW-PSNP beneficiary and eligible non-beneficiaryhouseholds.

Outcome Variables for Asset Accumulation: The first is the home asset (which is the sum of

household goods and consumer durables) and community asset value is also measured based on

the estimation of its market value by respondents in terms of Birr.

Lines 306 to 310 Therefore, in the study, the households have been asked about the quantity as well as the current price of overall households mean real food consumption expenditure per Adult Equivalent (AE) in terms of 24 hours and current non-food consumption expenditure per adult equivalent (AE) in terms of year, at their disposal, during the field survey based on the respondents’ consumption rate in terms of Ethiopian Birr

Line 311 3. Results and Discussion

There was no limitation section in this study, please kindly put this in place and also recommend /suggest future study/ research area for further studies.

**Go through the references and update as recommended by the article’s guidelines**

**7. PLOS authors have the option to publish the peer review history of their article (what does this mean?). If published, this will include your full peer review and any attached files**.

**Reviewer #1: No**

**Reviewer #3: No**

****

**While revising your submission, please upload your figure files to the Preflight Analysis and Conversion Engine (PACE) digital diagnostic tool, https://pacev2.apexcovantage.com/. PACE helps ensure that figures meet PLOS requirements. To use PACE, you must first register as a user. Registration is free. Then, login and navigate to the UPLOAD tab, where you will find detailed instructions on how to use the tool. If you encounter any issues or have any questions when using PACE, please email PLOS at figures@plos.org. Please note that Supporting Information files do not need this step.**

---

## [Author Response · Author response to Decision Letter 1]

12 Jun 2024

Based on the questions raised by reviewers, I have responded one by one as follow:

Author’s Response

Question 1. In the abstract, the authors should do some grammar checks on some tenses (appropriate use of past tense). And indicate the knowledge gap and/ problem statement.

Response: I made the necessary grammar correction and the inclusion of the knowledge gap.

Question 2: Lines 67 & 68 - An estimated 149 million Africans are facing acute food insecurity—an increase of 12 million people from a year ago. Is this a recent information? If yes please indicate the year of reference and source of the information with appropriate referencing.

Response: I include the following reference: 

Food and Agriculture Organization of the United Nations. (2023). The State of Food Security and Nutrition in the World 2023. Rome: FAO. 

Question 3: Lines 78 & 79 - Before implementing PSNP, the Ethiopian government relied on unpredictable annual appeals for emergency food relief, which saved lives but failed to address the increasing number of food insecure individuals [6].

Response: Numbers 69 and 70 are deleted because they were written mistakenly.

Question 4: Lines 84 to 89 - The study suggests that the PSNP plays a crucial role in ensuring food security for both chronically and temporarily food insecure households. By providing program transfers and supporting productive activities, the PSNP aims to assist over 7.5 million people in 318 districts across eight regions. This targeted approach is likely more effective in addressing the needs of food insecure individuals compared to the unpredictable annual appeals for emergency food relief. No reference to support this fact.

Response: I include the following reference:

 Gilligan, D. O., Hoddinott, J., & Seyoum Taffesse, A. (2008). The impact of Ethiopia's Productive Safety Net Programme and its linkages. International Food Policy Research Institute (IFPRI).

Question 5: Lines 90 to 93 - Unlike the initial emergency relief appeal method, which relied on unpredictable annual appeals for food aid, the PSNP operates on a set timeline until December 2014. This longer-term planning allows for more stability and predictability in addressing food insecurity issues. No reference.

Response: I include the following reference: 

Berhane, G., Gilligan, D. O., Hoddinott, J., Kumar, N., & Taffesse, A. S. (2014). Can social protection work in Africa? The impact of Ethiopia's Productive Safety Net Programme. Economic Development and Cultural Change, 63(1), 1-26.

Question 6: Lines 126 & 127 - Hence, Dessie City has been labeled as typical food insecure area despite various food security interventions with PSNP is effective or not. Please paraphrase the following statement:

Response: The statement is paraphrased as follow:

Despite the introduction of multiple food security interventions with PSNP, Dessie City has been classified as a typical food insecure area.

Question 7: Lines 134 &135 - Historically, urban poverty has consistently faced the significant challenge of food insecurity, especially in drought-prone areas of Ethiopia. No reference.

Response: I include the following reference: 

Zewdie, Yilma Muluken. 2014. "Food Insecurity and Coping Strategies of Urban Poor Households in Addis Ababa." Journal of Economics and Sustainable Development 5(20): 20-30.

Question 8: Lines 296-302 - Impact Indicator Variables for Asset Accumulation and consumption

The impact of the PSNP is analyzed by dividing the two categories which are the asset building and consumption expenditure of the PW-PSNP beneficiary and eligible non-beneficiary households. Reference for this asersion.

Response: I include the following reference: Gebrehiwot, T. and Castilla, C. 2019. "The Role of the Productive Safety Net Programme (PSNP) in Improving Consumption and Asset Accumulation in Rural Ethiopia." Journal of Development Studies 55(5): 685-700.

Question 9: Outcome Variables for Asset Accumulation: The first is the home asset (which is the sum of household goods and consumer durables) and community asset value is also measured based on the estimation of its market value by respondents in terms of Birr. Reference for this tool/ascersion.

Response: I include the following reference: Gilligan, Daniel O., and Hoddinott, John. 2007. "Is There Persistence in the Impact of Emergency Food Aid? Evidence on Consumption, Food Security, and Assets in Rural Ethiopia." American Journal of Agricultural Economics 89(2): 225-242.

Question 10: Lines 306 to 310 - Therefore, in the study, the households have been asked about the quantity as well as the current price of overall households mean real food consumption expenditure per Adult Equivalent (AE) in terms of 24 hours and current non-food consumption expenditure per adult equivalent (AE) in terms of year, at their disposal, during the field survey based on the respondents’ consumption rate in terms of Ethiopian Birr. Reference for the source of this measure/tool?

Response: I include the following reference: Gebrehiwot, Tesfaye, and Van der Veen, Anne. 2015. "Coping with Food Insecurity on a Micro-scale: Evidence from Ethiopian Rural Households." Ecology of Food and Nutrition 54(2): 187-208.

---

## [Decision Letter · Decision Letter 2]

25 Jun 2024

PONE-D-23-24451R2Impact of Urban Productive Safety Net Program on Urban Households’ Asset Accumulation and Food Consumption Rate in Dessie City, South Wollo Zone, Amhara Region, EthiopiaPLOS ONE

Dear Dr. Assefa,

Thank you for submitting your manuscript to PLOS ONE. After careful consideration, we feel that it has merit but does not fully meet PLOS ONE’s publication criteria as it currently stands. Therefore, we invite you to submit a revised version of the manuscript that addresses the points raised during the review process.

**ACADEMIC EDITOR: Experts in the field have reviewed your manuscript and you are expected to address their comments as early as possible. Thank you**.

**Please submit your revised manuscript by **Aug 09 2024 11:59PM**. If you will need more time than this to complete your revisions, please reply to this message or contact the journal office at plosone@plos.org. When you're ready to submit your revision, log on to https://www.editorialmanager.com/pone/ and select the 'Submissions Needing Revision' folder to locate your manuscript file**.

**Please include the following items when submitting your revised manuscript:****A rebuttal letter that responds to each point raised by the academic editor and reviewer(s). You should upload this letter as a separate file labeled 'Response to Reviewers'.****A marked-up copy of your manuscript that highlights changes made to the original version. You should upload this as a separate file labeled 'Revised Manuscript with Track Changes'.****An unmarked version of your revised paper without tracked changes. You should upload this as a separate file labeled 'Manuscript'.****If applicable, we recommend that you deposit your laboratory protocols in protocols.io to enhance the reproducibility of your results. Protocols.io assigns your protocol its own identifier (DOI) so that it can be cited independently in the future. For instructions see: https://journals.plos.org/plosone/s/submission-guidelines#loc-laboratory-protocols. Additionally, PLOS ONE offers an option for publishing peer-reviewed Lab Protocol articles, which describe protocols hosted on protocols.io. Read more information on sharing protocols at https://plos.org/protocols?utm_medium=editorial-email&utm_source=authorletters&utm_campaign=protocols.

**We look forward to receiving your revised manuscript.**

**Kind regards,**

**Olutosin Ademola Otekunrin**

**Academic Editor**

**PLOS ONE**

**Journal Requirements**:

Reviewers' comments:

**Reviewer's Responses to Questions**

**Comments to the Author**

**1. If the authors have adequately addressed your comments raised in a previous round of review and you feel that this manuscript is now acceptable for publication, you may indicate that here to bypass the “Comments to the Author” section, enter your conflict of interest statement in the “Confidential to Editor” section, and submit your "Accept" recommendation.**

**Reviewer #1: All comments have been addressed**

**Reviewer #3: (No Response)**

**2. Is the manuscript technically sound, and do the data support the conclusions?**

**The manuscript must describe a technically sound piece of scientific research with data that supports the conclusions. Experiments must have been conducted rigorously, with appropriate controls, replication, and sample sizes. The conclusions must be drawn appropriately based on the data presented. **

**Reviewer #1: Yes**

**Reviewer #3: Yes**

**3. Has the statistical analysis been performed appropriately and rigorously? **

**Reviewer #1: Yes**

**Reviewer #3: Yes**

**4. Have the authors made all data underlying the findings in their manuscript fully available?**

**The PLOS Data policy requires authors to make all data underlying the findings described in their manuscript fully available without restriction, with rare exception (please refer to the Data Availability Statement in the manuscript PDF file). The data should be provided as part of the manuscript or its supporting information, or deposited to a public repository. For example, in addition to summary statistics, the data points behind means, medians and variance measures should be available. If there are restrictions on publicly sharing data—e.g. participant privacy or use of data from a third party—those must be specified.**

**Reviewer #1: Yes**

**Reviewer #3: Yes**

**5. Is the manuscript presented in an intelligible fashion and written in standard English?**

**PLOS ONE does not copyedit accepted manuscripts, so the language in submitted articles must be clear, correct, and unambiguous. Any typographical or grammatical errors should be corrected at revision, so please note any specific errors here.**

**Reviewer #1: Yes**

**Reviewer #3: Yes**

**6. Review Comments to the Author**

**Please use the space provided to explain your answers to the questions above. You may also include additional comments for the author, including concerns about dual publication, research ethics, or publication ethics. (Please upload your review as an attachment if it exceeds 20,000 characters)**

**Reviewer #1: No additional comments. All comments have been taken into account**.

The paper can be published.

**Thank you**

**Reviewer #3: looks like the author wrongly uploaded an incorrect manuscript which does not tally with the corrections indicated in the feedback note attached by the authors. the authors need to kindly check up and ensure the corrected version is submitted subsequently.**

**7. PLOS authors have the option to publish the peer review history of their article (what does this mean?). If published, this will include your full peer review and any attached files**.

**Reviewer #1: No**

**Reviewer #3: No**

****

**While revising your submission, please upload your figure files to the Preflight Analysis and Conversion Engine (PACE) digital diagnostic tool, https://pacev2.apexcovantage.com/. PACE helps ensure that figures meet PLOS requirements. To use PACE, you must first register as a user. Registration is free. Then, login and navigate to the UPLOAD tab, where you will find detailed instructions on how to use the tool. If you encounter any issues or have any questions when using PACE, please email PLOS at figures@plos.org. Please note that Supporting Information files do not need this step.**

---

## [Author Response · Author response to Decision Letter 2]

2 Jul 2024

Based on the questions raised by reviewers, I have responded one by one as follow:

Author’s Response

Question 1. In the abstract, the authors should do some grammar checks on some tenses (appropriate use of past tense). And indicate the knowledge gap and/ problem statement.

Response: I made the necessary grammar correction and the inclusion of the knowledge gap.

Question 2: Lines 67 & 68 - An estimated 149 million Africans are facing acute food insecurity—an increase of 12 million people from a year ago. Is this a recent information? If yes please indicate the year of reference and source of the information with appropriate referencing.

Response: I include the following reference: 

Food and Agriculture Organization of the United Nations. (2023). The State of Food Security and Nutrition in the World 2023. Rome: FAO. 

Question 3: Lines 78 & 79 - Before implementing PSNP, the Ethiopian government relied on unpredictable annual appeals for emergency food relief, which saved lives but failed to address the increasing number of food insecure individuals [6].

Response: Numbers 69 and 70 are deleted because they were written mistakenly.

Question 4: Lines 84 to 89 - The study suggests that the PSNP plays a crucial role in ensuring food security for both chronically and temporarily food insecure households. By providing program transfers and supporting productive activities, the PSNP aims to assist over 7.5 million people in 318 districts across eight regions. This targeted approach is likely more effective in addressing the needs of food insecure individuals compared to the unpredictable annual appeals for emergency food relief. No reference to support this fact.

Response: I include the following reference:

 Gilligan, D. O., Hoddinott, J., & Seyoum Taffesse, A. (2008). The impact of Ethiopia's Productive Safety Net Programme and its linkages. International Food Policy Research Institute (IFPRI).

Question 5: Lines 90 to 93 - Unlike the initial emergency relief appeal method, which relied on unpredictable annual appeals for food aid, the PSNP operates on a set timeline until December 2014. This longer-term planning allows for more stability and predictability in addressing food insecurity issues. No reference.

Response: I include the following reference: 

Berhane, G., Gilligan, D. O., Hoddinott, J., Kumar, N., & Taffesse, A. S. (2014). Can social protection work in Africa? The impact of Ethiopia's Productive Safety Net Programme. Economic Development and Cultural Change, 63(1), 1-26.

Question 6: Lines 126 & 127 - Hence, Dessie City has been labeled as typical food insecure area despite various food security interventions with PSNP is effective or not. Please paraphrase the following statement:

Response: The statement is paraphrased as follow:

Despite the introduction of multiple food security interventions with PSNP, Dessie City has been classified as a typical food insecure area.

Question 7: Lines 134 &135 - Historically, urban poverty has consistently faced the significant challenge of food insecurity, especially in drought-prone areas of Ethiopia. No reference.

Response: I include the following reference: 

Zewdie, Yilma Muluken. 2014. "Food Insecurity and Coping Strategies of Urban Poor Households in Addis Ababa." Journal of Economics and Sustainable Development 5(20): 20-30.

Question 8: Lines 296-302 - Impact Indicator Variables for Asset Accumulation and consumption

The impact of the PSNP is analyzed by dividing the two categories which are the asset building and consumption expenditure of the PW-PSNP beneficiary and eligible non-beneficiary households. Reference for this asersion.

Response: I include the following reference: 

Gebrehiwot, T. and Castilla, C. 2019. "The Role of the Productive Safety Net Programme (PSNP) in Improving Consumption and Asset Accumulation in Rural Ethiopia." Journal of Development Studies 55(5): 685-700.

Question 9: Outcome Variables for Asset Accumulation: The first is the home asset (which is the sum of household goods and consumer durables) and community asset value is also measured based on the estimation of its market value by respondents in terms of Birr. Reference for this tool/ascersion.

Response: I include the following reference: 

Gilligan, Daniel O., and Hoddinott, John. 2007. "Is There Persistence in the Impact of Emergency Food Aid? Evidence on Consumption, Food Security, and Assets in Rural Ethiopia." American Journal of Agricultural Economics 89(2): 225-242.

Question 10: Lines 306 to 310 - Therefore, in the study, the households have been asked about the quantity as well as the current price of overall households mean real food consumption expenditure per Adult Equivalent (AE) in terms of 24 hours and current non-food consumption expenditure per adult equivalent (AE) in terms of year, at their disposal, during the field survey based on the respondents’ consumption rate in terms of Ethiopian Birr. Reference for the source of this measure/tool?

Response: I include the following reference: 

Gebrehiwot, Tesfaye, and Van der Veen, Anne. 2015. "Coping with Food Insecurity on a Micro-scale: Evidence from Ethiopian Rural Households." Ecology of Food and Nutrition 54(2): 187-208.

---

## [Decision Letter · Decision Letter 3]

26 Jul 2024

Impact of Urban Productive Safety Net Program on Urban Households’ Asset Accumulation and Food Consumption Rate in Dessie City, South Wollo Zone, Amhara Region, Ethiopia

PONE-D-23-24451R3

Dear Dr. Assefa,

We’re pleased to inform you that your manuscript has been judged scientifically suitable for publication and will be formally accepted for publication once it meets all outstanding technical requirements.

Kind regards,

Olutosin Ademola Otekunrin

Academic Editor

PLOS ONE

Additional Editor Comments (optional):

Reviewers' comments:

Reviewer's Responses to Questions

**Comments to the Author**

1. If the authors have adequately addressed your comments raised in a previous round of review and you feel that this manuscript is now acceptable for publication, you may indicate that here to bypass the “Comments to the Author” section, enter your conflict of interest statement in the “Confidential to Editor” section, and submit your "Accept" recommendation.

Reviewer #3: All comments have been addressed

2. Is the manuscript technically sound, and do the data support the conclusions?

Reviewer #3: Yes

3. Has the statistical analysis been performed appropriately and rigorously? 

Reviewer #3: Yes

4. Have the authors made all data underlying the findings in their manuscript fully available?

Reviewer #3: Yes

5. Is the manuscript presented in an intelligible fashion and written in standard English?

Reviewer #3: Yes

6. Review Comments to the Author

Reviewer #3: The authors have done incredibly well to painstakingly address all the concerns raised on this subject matter, and I wish to congratulate them for a job well done. The reviewer is satisfied with the work done and wishes to recommend that the manuscript proceed to the publication stage.

7. PLOS authors have the option to publish the peer review history of their article (what does this mean?). If published, this will include your full peer review and any attached files.

Reviewer #3: No

---

## [Editor Report · Acceptance letter]

6 Aug 2024

PONE-D-23-24451R3 

PLOS ONE

Dear Dr. Assefa, 

I'm pleased to inform you that your manuscript has been deemed suitable for publication in PLOS ONE. Congratulations! Your manuscript is now being handed over to our production team.

Kind regards, 

on behalf of

Dr. Olutosin Ademola Otekunrin 

Academic Editor

PLOS ONE